# Differences in sensory nerve block between levobupivacaine and bupivacaine at low concentrations in humans and animals

**Akiyuki Sakamoto**[1,2]*, **Satoshi Tanaka**[1], **Takashi Ishida**[1], **Mikito Kawamata**[1]

1 Department of Anesthesiology and Resusitology, Shinshu University School of Medicine, Matsumoto, Japan, 2 Department of Anesthesiology, Shinonoi General Hospital, Nagano, Japan

* akiyuki@shinonoi-hp.jp

## Abstract

Physiochemical properties of levobupivacaine and bupivacaine are identical, but pharmacokinetic and pharmacodynamics properties differ due to stereoselective interactions at the molecular sites of action. An evaluation of nerve block characteristics is essential for optimal clinical application. This study compared the sensory blocking characteristics of levobupivacaine to bupivacaine in humans and model animals. Levobupivacaine and bupivacaine were compared for sensory block efficacy using a randomized, double-blinded, crossover study design. Eighteen healthy volunteers were randomized to receive levobupivacaine or bupivacaine by subcutaneous injection into the forearm, followed by the other drug 1 week later with injection order counterbalanced across subjects. Tactile detection and mechanical pain thresholds were determined using von Frey hairs and thermal pain threshold using a thermal stimulator. Effects of levobupivacaine and bupivacaine, on the spiking activity of spinal dorsal horn (SDH) neurons evoked by innocuous or noxious stimuli were also compared in anesthetized Sprague–Dawley rats by in vivo extracellular recordings. There were no significant differences in mechanical and thermal pain thresholds following levobupivacaine or bupivacaine injection at 0.025%, 0.0625%, and 0.125%. There was also no significant difference in tactile detection threshold following levobupivacaine or bupivacaine injection at 0.125%. However, tactile detection threshold was significantly higher after administration of bupivacaine at 0.025% and 0.0625% compared to equivalent doses of levobupivacaine. Subcutaneous injection of bupivacaine at 0.05% also induced significantly greater inhibition of SDH neuron spiking activity evoked by innocuous stimuli compared to an equivalent dose of levobupivacaine, while there was no significant difference in suppression of spiking activity evoked by noxious stimuli. Low-dose bupivacaine induces greater suppression tactile sensation than low-dose levobupivacaine. Thus, low-dose levobupivacaine demonstrates relatively greater blocking selectivity for noxious over innocuous stimuli compared to low-dose bupivacaine. Levobupivacaine may be advantageous for applications where pain must be suppressed but non-nociceptive sensations maintained.

**Data availability statement:** All relevant data are within the article and its Supporting Information files.

**Funding:** This work was financially supported by the Japan Society for the Promotion of Science Grants-in-Aid for Scientific Research Grant Number 23791696. The funders had no role in study design, data collection and analysis, decision to publish, or preparation of the manuscript.

**Competing interests:** The authors have declared that no competing interests exist.

## Introduction

Bupivacaine is a long-acting local anesthetic that has been used clinically for several decades as a racemic mixture (50:50) of dextrorotatory R-(+)- and levorotatory S-(−)-isomers. However, the R-(+)-isomer may contribute disproportionately to adverse effects on the nervous and cardiovascular systems compare to the S-(−)-isomer [1–5]. Therefore, levobupivacaine, the pure S-(−)-isomer of bupivacaine, has been introduced into clinical practice as a safer alternative. Although the physiochemical properties of levobupivacaine and bupivacaine are identical, they differ in pharmacokinetic and pharmacodynamics properties due to stereoselective interactions at molecular targets [4–6].

Local anesthetics at sufficient concentrations can prevent impulse transmission by all types of peripheral nerves, resulting in motor blockade and sensory blockade for both noxious and innocuous stimuli. Complete block of motor and sensory transmission is generally beneficial during surgery, while nociceptive-specific block without motor paralysis or loss of innocuous sensation is often desirable in the postoperative period to facilitate earlier mobilization. The motor-blocking potency of levobupivacaine is lower than that of bupivacaine at the same concentration and amount when administered by intrathecal and epidural routes [5,7–10]. Recently, Uta and colleagues reported that levobupivacaine also potently inhibits Aδ- and C-fiber transmission but requires a higher dose to suppress Aβ fiber transmission compared to bupivacaine as evidenced by whole cell patch-clamp recordings from rat spinal dorsal horn (SDH) neurons [11]. Nerve fibers of the Aβ-type transmit tactile and pressure sensations, whereas Aδ- and C-fibers transmit nociception from noxious stimuli [12]. Therefore, we hypothesized that levobupivacaine would produce greater nociceptive-specific block than bupivacaine, a property especially useful in the outpatient surgery setting. However, these differential blocking properties have not been confirmed in vivo.

It is of great importance to characterize the sensory block characteristics of bupivacaine and levobupivacaine for optimal clinical applications. The purpose of the present study was to compare the relatively selectivity of racemic bupivacaine to levobupivacaine for pain sensation and tactile sensation block by measuring sensory thresholds in human volunteers and the spiking activity of SDH neurons in response to noxious and innocuous cutaneous stimuli by in vivo single unit extracellular recordings in anesthetized rats.

## Materials and methods

This study consists of two parts. In Experiment (1), we compared the effects of levobupivacaine to bupivacaine on the tactile detection threshold and both mechanical and thermal pain sensation thresholds following subcutaneous administration in healthy human volunteers. In Experiment (2), the effects of subcutaneous levobupivacaine and bupivacaine administration were compared for suppression of SDH neuron activity evoked by innocuous and noxious stimuli in anesthetized rats.

### Experiment (1): Effects of subcutaneous levobupivacaine and bupivacaine on human sensory thresholds

**Subjects.** Experiment (1) was conducted in accordance with the principles outlined in the Declaration of Helsinki and the ethical guidelines for pain research in humans of the International Association for Study of Pain. All protocols were approved by the Ethics Committee of Shinshu University School of Medicine, Matsumoto, Japan (document number: 3252) on October 8, 2015. Experiment (1) was registered with the University Hospital Medical Information Network in Japan (number UMIN000019307) on October 10, 2015. Study participants were recruited between October 13, 2015, and February 15, 2018. Data

were collected from October 18, 2015, to March 3, 2018, at Shinshu University Hospital in Matsumoto, Japan. Written informed consent was obtained from each subject before testing. A randomized, double-blinded, crossover design was implemented to compare different subcutaneous doses of levobupivacaine and bupivacaine without bias. Eighteen volunteers (15 males and 3 females) aged 23–46 were recruited at Shinshu University School of Medicine. Inclusion criteria were 20 years of age or older and willing to provide informed consent. Exclusion criteria were analgesic allergies, drug or alcohol abuse, diabetes, neuromuscular diseases, chronic pain, and daily use of analgesics.

**Study protocol.** A CONSORT diagram of study enrolment is presented in Fig 1. The 18 volunteers were randomly assigned to receive either levobupivacaine in the first period followed by bupivacaine in the second period or bupivacaine in the first period followed by levobupivacaine in the second period at a 1:1 ratio (crossover design). Volunteers were randomized to the levobupivacaine-first or bupivacaine-first group using the random

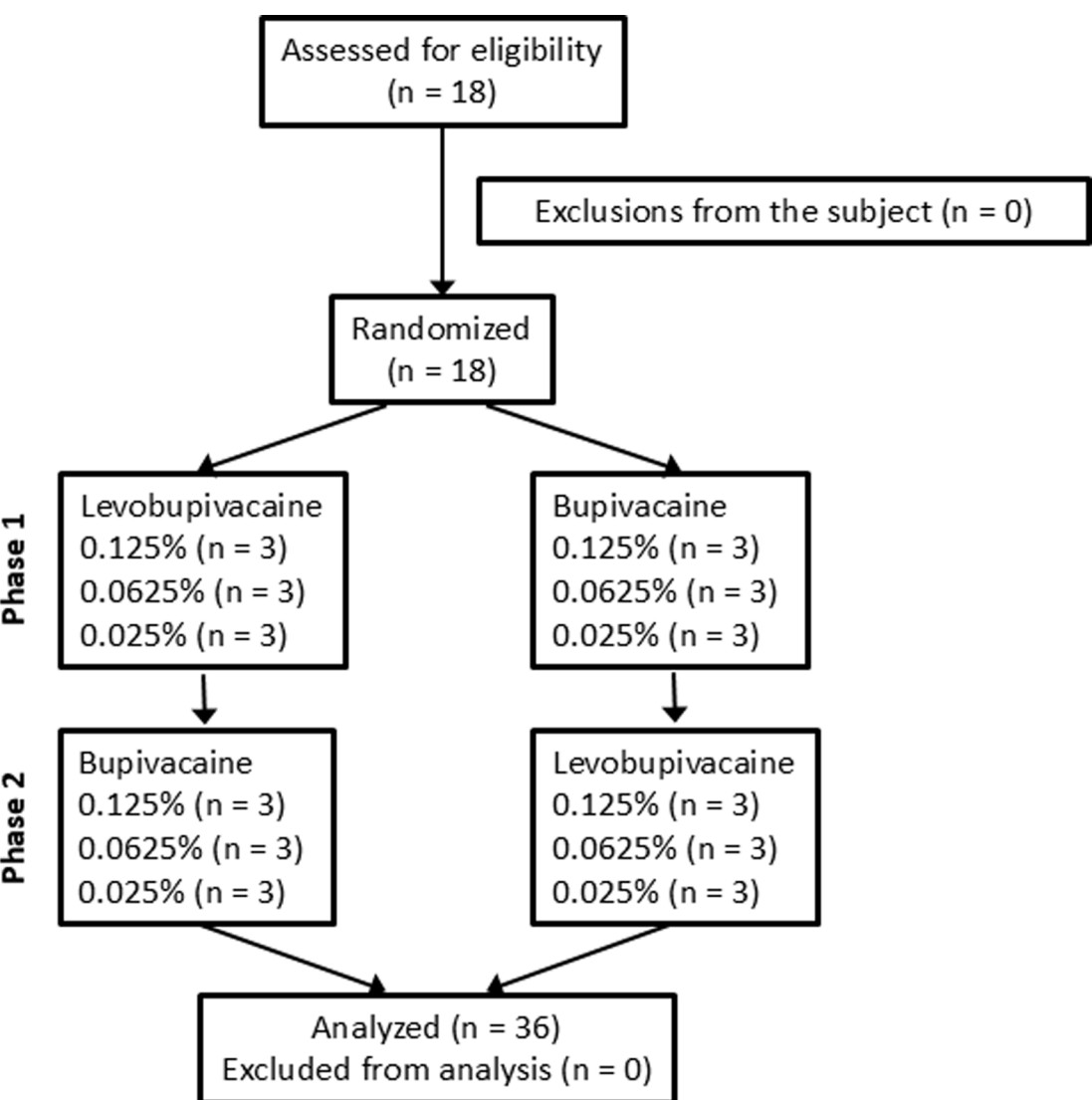

**Fig 1. CONSORT diagram illustrating the flow of participants through each phase of the randomized crossover trial.**

generation function in Microsoft Excel with a permuted random block size of 6. Subjects were further randomly divided into three dose groups receiving 0.125%, 0.0625%, or 0.025% injections. These groups were administered the same dose of the other local anesthetic after a 1-week washout period. The commercial 0.5% bupivacaine preparation Marcaine® (AstraZeneca KK, Osaka, Japan) and the 0.5% levobupivacaine preparation Popscaine® (Maruishi, Tokyo, Japan) were diluted in sterile normal saline on the test day to yielded the 0.125%, 0.0625%, or 0.025% solutions for subcutaneous injection. Freshly prepared solutions were injected into the anterior aspect of the left forearm at a volume of 3 mL using a 25-gauge needle under ultrasound guidance (S-Nerve™, SonoSite Japan KK., Tokyo, Japan). The tactile detection threshold (TDT) and mechanical pain threshold (MPT) were determined at the center of the injected area by applying von Frey (vF) hairs of increasing stiffness (force). The TDT was measured from a baseline force of 0.16 g and MPT from a baseline of 6 g. Participants were instructed to announce when touch was first detected (TDT) and when the stimulus became painful (MPT). Each threshold was measured three times at intervals of 10 s, and the median value was recorded for analyses. The thermal pain threshold (TPT) was measured using a thermal stimulator (THERMAL STIMULATOR®, Dia-medical Co., Tokyo, Japan). The thermode, a Peltier element covered by a ceramic contact plate (6 mm × 6 mm), was heated at a rate of 1.0 °C/s from a baseline temperature of 32 °C to a maximum of 47 °C to prevent injury. Participants were instructed to notify when thermal pain sensation was detected. Evaluators of these sensory thresholds were blinded to the anesthetic administered. Measurements were performed before injection and 5, 15, 30, 45, and 60 min after injection in a quiet room with controlled ambient temperature (22 °C–24 °C) (Fig 2).

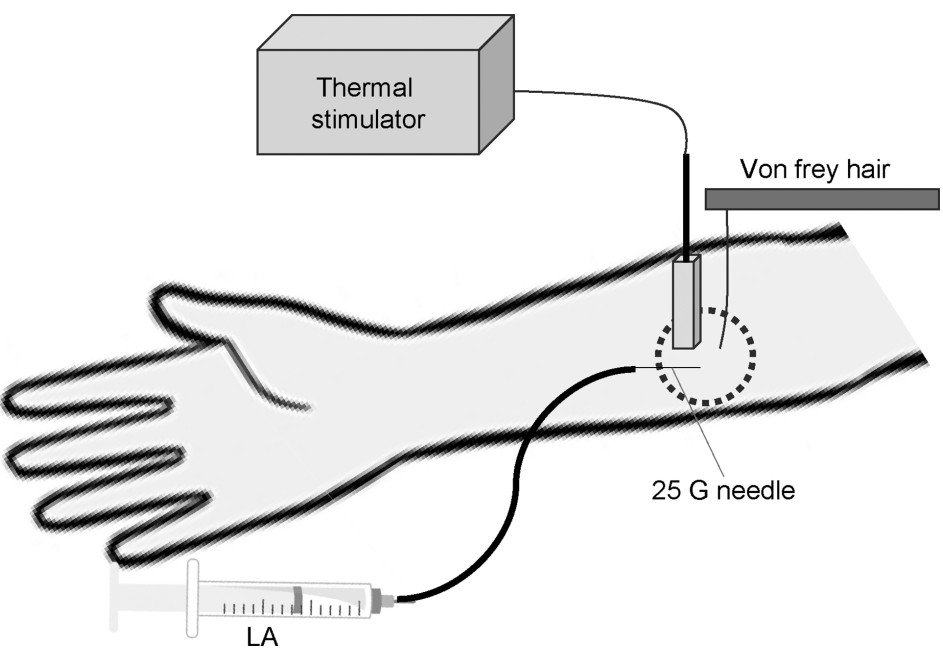

**Fig 2. Schematic diagram of Experiment (1) in humans.** Tactile and mechanical pain thresholds were determined using von Frey (vF) hairs applied for 5 s, while thermal pain threshold was measured using a thermal stimulator (THERMAL STIMULATOR®, Dia-medical Co., Tokyo, Japan) heated at 1 °C/min from 32 °C to a maximum of 47 °C. All thresholds were measured at baseline and again at the same site and times following subcutaneous injection of local anesthetic (LA).

To control for possible carryover effects from one injection to the next, the total scores for each period were compared between two sequence groups using an independent samples t-test. Furthermore, the effect of each period was examined by comparing the thresholds in the first period with the thresholds in the second period using paired-sample t-tests [13].

## Experiment (2): Effects of subcutaneous levobupivacaine and bupivacaine on spinal dorsal horn transmission in anesthetized rats

All animal care and study protocols were approved by the Shinshu University School of Medicine Animal Care and Use Committee (no. 021118) and conducted in accordance with the guidelines of the National Institutes of Health. Seven-week-old male Sprague–Dawley rats weighing 180–220 g were housed under controlled temperature (22–23 °C), humidity 40%–60%, and light/dark cycle (12-hour/12-hour) with ad libitum access to food and water.

Prior to neuronal recordings, rats were anesthetized with 3% sevoflurane in oxygen. A middle vertical incision was made over the dorsum and the underlying paraspinous musculature was detached from the spinous processes and dorsal aspects of vertebrae from T10 to L3. Dorsal laminectomies were performed across T12–L1 to expose the lumber intumescence. The rat was then placed into a stereotaxic frame (Model ST-7, Narishige, Tokyo, Japan) and secured using vertebral clamps and ear bars. After removing the dura, the surface of the spinal cord was irrigated at 10 mL/min with Kreb's solution (in mM, NaCl, 117; KCl, 3.6; $CaCl_2$, 1.2; $NaH_2PO_4$, 1.2; glucose, 11; $NaHCO_3$, 25 mM) aerated with 95% $O_2$–5% $CO_2$. Body temperature was maintained at 36ºC–38ºC using an infrared heat lamp and thermo-controlled heat pad based on feedback from a rectal thermometer.

Extracellular recordings from single neurons were acquired using a tungsten electrode (10–12 MΩ; FHC Inc., Brunswick, ME) inserted in the deep dorsal horn of the lumbar spinal cord. We identified neurons (single units) according to the mechanical receptive fields (RF) of the hindpaw. Extracellular action potentials were amplified (×20,000–50,000), band-pass filtered between 300–3,000 Hz, digitized (CED 1401; Cambridge Electronic Design, Cambridge, UK), and stored on an IBM-AT personal computer (Think Pad; IBM Japan, Tokyo). Spike trains were analyzed using Spike 2 (Cambridge Electronic Design).

Neurons that responded to both an innocuous stimulus (light touch with a camel-hair brush) and a noxious stimulus (pinch force of 250 g/mm$^2$ using an arterial clip) were classified as wide-dynamic-range (WDR), those responsive to noxious but not non-noxious stimuli as high-threshold, and those responsive to low- but not high-intensity stimulation as low-threshold (LT). Only WDR and LT neurons were examined in this study. Consistent with previous reports, all WDR neurons included in the analysis responded to greater stimulus intensities with a graded increase in spike frequency [14], while LT neurons responded only to light mechanical (innocuous) stimuli from a 10 g or lighter vF hair [15]. Therefore, the pinch stimulus was used throughout as a strong noxious stimulus and a 10 g vF hair as a weak noxious stimulus. The responses of individual WDR neurons to stimulation by 4 g and 10 g vF hairs, a brush, and pinch at the center of the RF were recorded sequentially. Similarly, the responses of LT neurons to stimulation by a 4 g vF hair and a brush at the center of the RF were recorded sequentially. For measurement of levobupivacaine and bupivacaine effects on these responses, rats were randomly allocated to receive a 500-microL subcutaneous injection of 0.05% levobupivacaine, 0.05% bupivacaine, or equal-volume saline using a 30-gauge needle targeted to the center of the RF. The responses to punctate mechanical stimulation using 4 or 10 g vF hairs with sufficient force to bend the hairs for 5 s were recorded. Additionally, the responses to one stroke from a camel-hair brush and a pinch stimulation from an arterial clip for 5 s were recorded. The neuronal responses to mechanical stimuli were recorded 5, 10, 20, 30, 60, and 90 min after injection by investigators blinded to group allocation

(levobupivacaine, bupivacaine, or saline). After completion of extracellular recording, all experimental animals were euthanized by inhalation of an excess amount of sevoflurane.

## Statistical analyses

We used G*Power 3.1 to determine the sample size, with a type I error rate (α) of 0.05, power value of 80% (1 − β = 0.80), an effect size of 2, and two-sided t-test. The minimum required group size for our human volunteer study was n = 5. Given the crossover design and potential dropouts, we increased the sample size to six. Our animal study established the sample size referring to prior studies that employed a similar design [16] and power analysis with G*Power software. Following the Shapiro–Wilk normality test, data were represented as mean ± standard deviation or median (25%, 75% interquartile range), depending on appropriateness. For within-group comparisons, Friedman and Dunn's post hoc tests were utilized, with p < 0.05 considered statistically significant. The Mann–Whitney U test was employed for group comparisons. Multiple testing indicates the increased risk of type I error while repeatedly conducting multiple statistical tests. The false discovery rate (FDR) correction aims to minimize the occurrence of type I errors at a manageable proportion [17]. The FDR-adjusted p-value was computed using this method, with an FDR-adjusted p-value < 0.05 deemed significant. All statistical analyses were performed using GraphPad Prism version 9.0 (GraphPad Software, San Diego, CA).

## Results

### Experiment (1): Subcutaneous injection of low-dose bupivacaine induced greater suppression of tactile sensation than low-dose levobupivacaine in human volunteers

**Patient characteristics.** All 18 healthy volunteers completed the study without major side effects. Demographic characteristics are summarized in Table 1.

**Carryover and period effects.** There were no significant differences in baseline TDT, MPT, and TPT, and no significant differences in threshold changes at 15 min postadministration between periods and anesthetics at equivalent doses (Supplementary Table S1), indicating that changes in TDT, MPT, and TPT induced by bupivacaine or levobupivacaine administration in the first period had disappeared by the start of the second period. Therefore, pooled data from the first and second periods were used to investigate the effects of levobupivacaine and bupivacaine on all outcomes.

**Effects of levobupivacaine and bupivacaine on sensory thresholds.** There were no significant differences in the TDT (Fig 3A), MPT (Fig 3B), and TPT (Fig 3C) at baseline between anesthetic groups. As expected, TDT increased significantly after subcutaneous administration of both levobupivacaine and bupivacaine at concentrations of 0.125%, 0.0625%, and 0.025%. There was no significant difference in TDT between anesthetic groups

Table 1. Demographic characteristics of human volunteers.

|  | Male | Female | Total |
|---|---|---|---|
| **N (%)** | 15 (83.3) | 3 (16.7) | 18 (100) |
| **Age (y.o.)** | 33 ± 7 | 27 ± 4 | 32 ± 7 |
| **Height (cm)** | 169 ± 6 | 155 ± 10 | 167 ± 9 |
| **Weight (kg)** | 65 ± 13 | 45 ± 9 | 61 ± 15 |

Data are presented as mean ± standard deviation

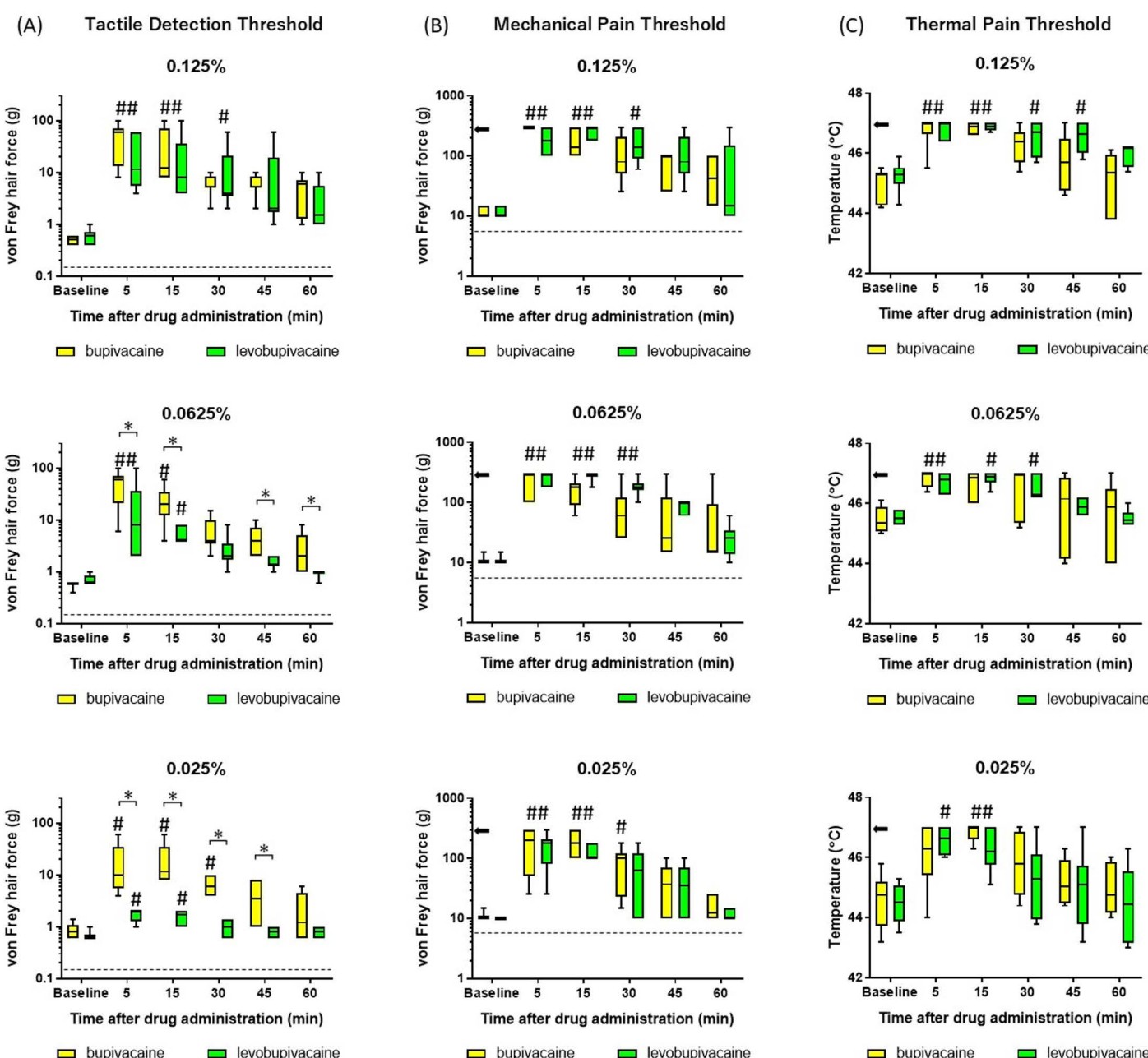

**Fig 3. Changes in tactile and pain thresholds in human volunteers following subcutaneous bupivacaine and levobupivacaine injections.** (A, B) Tactile detection threshold (TDT) (A) and mechanical pain threshold (MPT) (B) measured using von Frey hairs. (C) Thermal pain threshold (TPT) measured by a thermal stimulator. The TDT was significantly higher (tactile response less sensitive) following bupivacaine injection at concentrations of 0.025% and 0.0625% compared to equal-dose levobupivacaine (A). At the same time, MPT and TPT did not differ between LAs at any concentrations tested (B, C). *Arrows* indicate the cutoffs (B, C), and the dotted line indicates the baseline level (A, B). The Friedman test, followed by Dunn's post hoc test, was utilized for within-group comparisons, and the Mann–Whitney U test was employed for between-group comparisons. FDR-adjusted p-values < 0.05 indicated significant between-group differences. #: significant differences compared to baseline. *: significant differences between bupivacaine and levobupivacaine.

at 0.125%. However, the TDT significantly increased at 5, 15, 45, and 60 minutes after subcutaneous injection of 0.0625% bupivacaine, compared to the same concentration of levobupivacaine at these same time points (FDR-adjusted p-values = 0.034 for all). Similarly, at 5-, 15-, 30-, and 45-min postadministration of 0.025% bupivacaine, TDT also showed

higher values relative to levobupivacaine at those times (FDR-adjusted p-values = 0.007, 0.007, 0.007, and 0.024 respectively). Both levobupivacaine and bupivacaine also significantly increased the MPT at all concentrations tested as quantified by mean vF hair force, and the TPT produced by contact heat at 5 min after injection compared to before injection, without significant differences between drugs at equivalent concentrations and postinjection times (Figs 3B and 3C). Numerical data are shown in Supplementary Table S2.

### Experiment (2): Subcutaneous injection of bupivacaine induced greater suppression of SDH neuron activity evoked by innocuous stimulation than equal-dose levobupivacaine in anesthetized rats

A total of 37 neurons were identified from the T13 to L1 SDH of 37 anesthetized rats, of which 19 were classified as WDR and the remaining 18 as LT neurons. Of these, complete baseline and postinjection spike recordings were obtained from 13 WDR and 13 LT neurons for analysis. The mean of depth of the electrode tip position below the dorsal spinal cord surface was 745 ± 103 μm for WDR neurons and 432 ± 70 μm for LT neurons, consistent with the known anatomic distribution.

**Effects of bupivacaine and levobupivacaine on WDR neuron activity evoked by innocuous and noxious stimuli.** Figs 4A and 5A show typical spike discharge patterns of WDR neurons in response to noxious stimuli (pinch and 10 g vF hair) and innocuous mechanical stimuli (brush and 4 g vF hair) within corresponding RF at baseline and 10, 30, and 60 min after subcutaneous administration of levobupivacaine or bupivacaine, respectively, while Figs 4B and 5B show the median percent change. Both levobupivacaine and bupivacaine at 0.05% significantly inhibited the spiking responses of WDR neurons induced by noxious stimuli without significant difference (Fig 4B). Bupivacaine's inhibitory effects on WDR neuron responses to innocuous brushing at 5–60 minutes and 4 g vF at 30 and 60 minutes were significantly greater than those of levobupivacaine (brushings at 5, 10, 20, 30, and 60 min, FDR-adjusted p-values = 0.025, 0.024, 0.024, 0.024, and 0.024, respectively; 4 g vF, 30 and 60 min, FDR-adjusted p-values = 0.026, respectively) (Fig 5B). This finding aligns with the effects of bupivacaine on tactile thresholds in humans. Numerical data are shown in Supplementary Table S3.

**Effects of bupivacaine and levobupivacaine on LT neuron activity in response to innocuous stimuli.** Fig 6A shows typical discharge patterns of LT neurons in response to innocuous mechanical stimuli (brush and 4 g vF hair) within the corresponding RF at baseline and 10, 30, and 60 min after subcutaneous administration of levobupivacaine or bupivacaine, while Fig 6B shows the mean changes. The mean inhibitory effects of bupivacaine on the responses of LT neurons to brush strokes at 30 and 60 min postinjection were significantly stronger than those of levobupivacaine (30 and 60 min, the FDR-adjusted p-values were 0.048 for both cases) (Fig 6B). Similarly, the inhibitory effects of bupivacaine on the responses of LT neurons to 4 g vF at 20 and 30 min postinjection were significantly stronger than those of levobupivacaine (20 and 30 min, the FDR-adjusted p-values were 0.024 for both cases) (Fig 6B), again consistent with the effects observed on tactile thresholds in humans.

## Discussion

The major findings of this study are as follows. (1) In human volunteers, both 0.125% bupivacaine and 0.125% levobupivacaine increased pain and tactile thresholds in response to vF hairs and heat stimuli, while 0.025% and 0.0625% levobupivacaine enhanced pain thresholds but not thresholds for detection of innocuous stimuli. In contrast, bupivacaine at the same low concentrations enhanced thresholds to both noxious and innocuous stimuli. In other

(A)

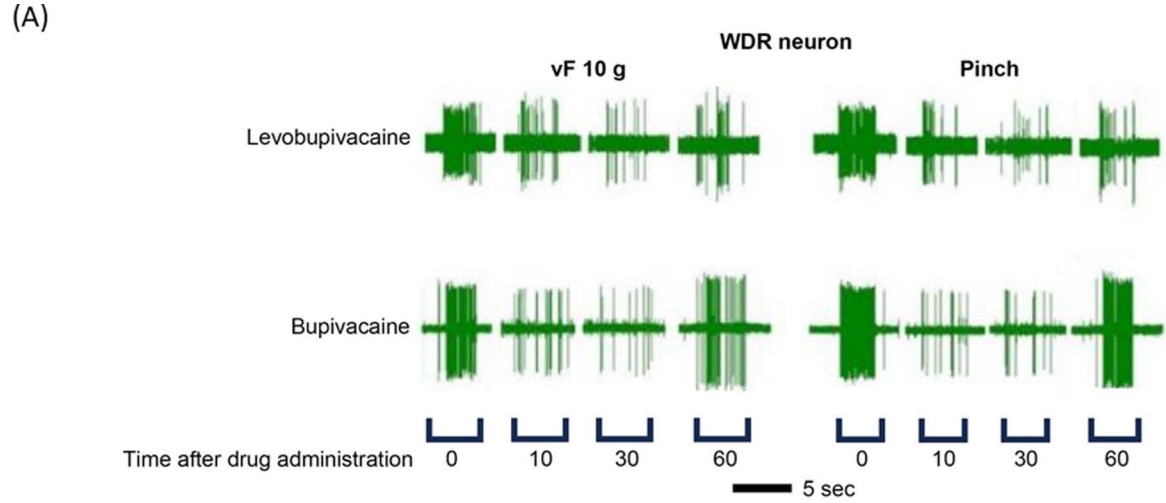

(B)

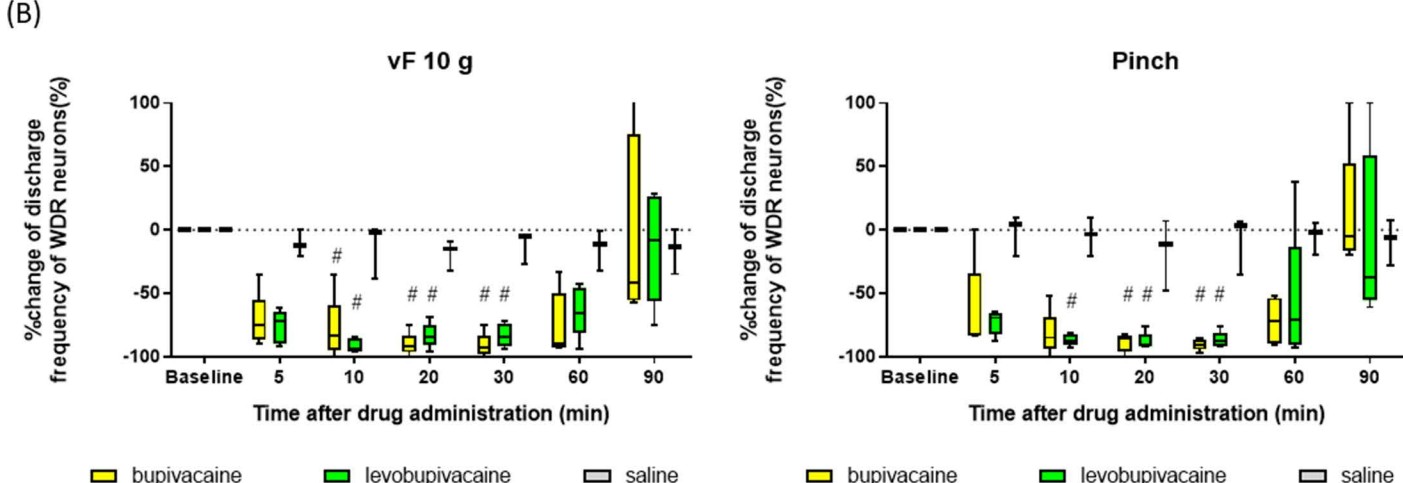

**Fig 4. Both bupivacaine and levobupivacaine strongly suppressed the spiking responses of wide dynamic-range neurons to noxious stimuli in anesthetized rats.** (A) Typical spike responses of wide-dynamic-range (WDR) neurons to noxious stimuli (pinch and 10 g von Frey hair) at baseline and after subcutaneous administration of bupivacaine or levobupivacaine (both 0.05%). (B) Values represent the median percentage change following subcutaneous administration of bupivacaine (n = 5), levobupivacaine (n = 5), or saline (n = 3), compared with the number of spiking responses to noxious stimuli at the preadministration baseline. Baseline responses across the groups were analyzed using the Friedman test and evaluated using Dunn's post hoc tests. The Mann–Whitney U test assessed changes induced by anesthesia between the two groups. An FDR-adjusted p-value of < 0.05 from the Mann–Whitney U test indicated significant differences between the groups. #: significant differences vs. baseline. *: significant differences between bupivacaine and levobupivacaine.

words, low-dose bupivacaine suppressed sensation to innocuous stimuli more powerfully than levobupivacaine. (2) In extracellular in vivo recordings from rat SDH neurons, both 0.05% bupivacaine and 0.05% levobupivacaine suppressed the spiking responses to noxious stimuli (from a 10 g vF hair and clamping) with equal efficacy similarly. Meanwhile, bupivacaine's inhibitory effects on the LT neurons' responses to 4 g vF were significantly more potent than those of levobupivacaine. To our knowledge, this is the first study demonstrating that low concentrations of bupivacaine and levobupivacaine have distinct anesthetic effects in humans, with levobupivacaine suppressing nociceptive mechanical and thermal sensations as effectively as bupivacaine but preserving innocuous tactile sensation. This differential effect

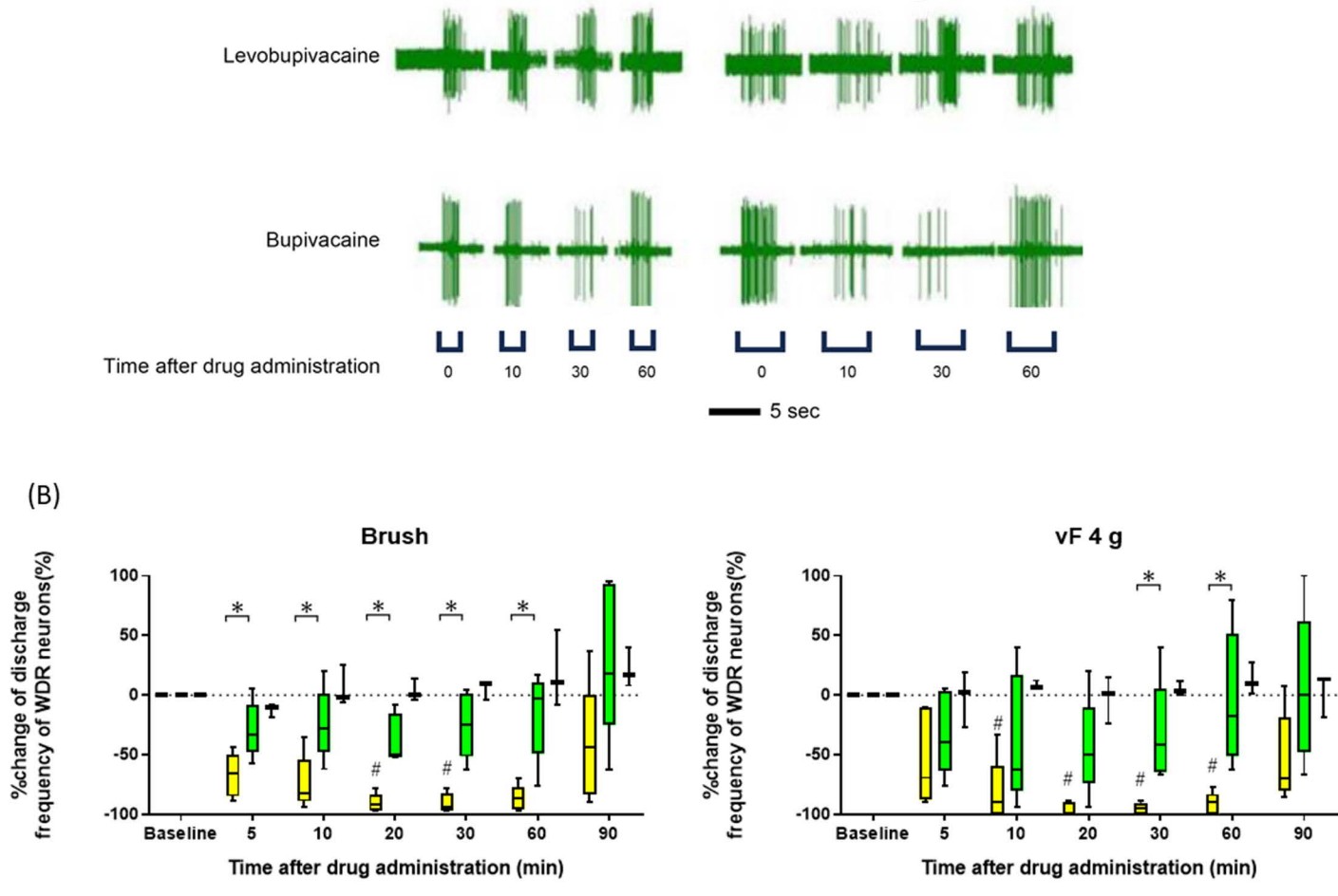

**Fig 5. Bupivacaine suppressed the spiking response of wide-dynamic-range neurons to innocuous stimuli more potently than levobupivacaine in anesthetized rats.** (A) Typical responses of wide-dynamic-range (WDR) neurons to innocuous stimuli (brush and 4 g von Frey hair) at baseline and after subcutaneous injection of bupivacaine and levobupivacaine (A). (B) Values represent the median percent change in spiking responses to innocuous stimuli following subcutaneous administration of bupivacaine (n = 5), levobupivacaine (n = 5), or saline (n = 3) compared with measurements taken before drug administration (baseline). The suppression of firing was stronger after administration of bupivacaine. The baseline responses of different groups were analyzed using Friedman and Dunn's post hoc tests. The Mann–Whitney U test assessed between-group differences in anesthesia-induced changes. A significant difference was identified when the FDR-adjusted p-value from the Mann–Whitney U test was < 0.05. #: significant differences vs. baseline. *: significant differences between bupivacaine and levobupivacaine.

on sensation in humans is in accord with the distinct effects of these two LAs on the spiking responses of rat SDH neurons to innocuous and noxious stimuli. Thus, this study provides a plausible mechanism for the differential effects of bupivacaine and levobupivacaine on tactile sensation and nociception, and further suggests that low-dose levobupivacaine can suppress pain as effectively as bupivacaine but without producing a numbing sensation that may delay postoperative mobilization.

Sensory threshold measurements in humans revealed no significant differences between bupivacaine and levobupivacaine at 0.125%, in accord with previous clinical studies reporting no significant differences in postoperative pain and motor block between bupivacaine and

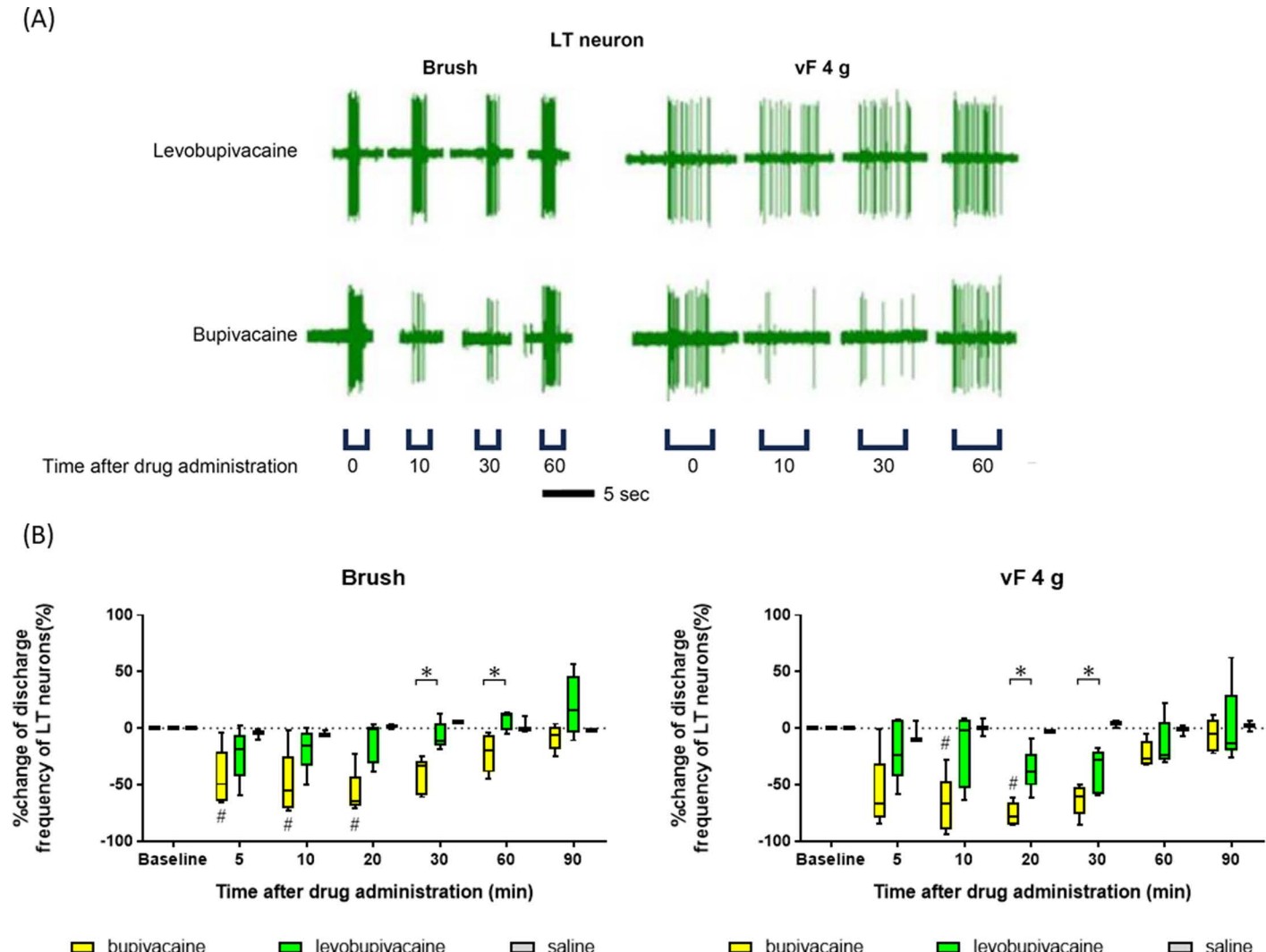

**Fig 6. Bupivacaine suppressed the spiking response of low-threshold neurons to innocuous stimuli more potently than levobupivacaine in anesthetized rats.** (A) Typical responses of low-threshold (LT) neurons to innocuous stimuli (brush and 4 g von Frey hair) at baseline and after subcutaneous administration of bupivacaine and levobupivacaine. (B) Values reflect the median percent changes after subcutaneous bupivacaine (n = 5), levobupivacaine (n = 5), or saline (n = 3) administration relative to the number of spiking responses to innocuous stimuli predrug administration (baseline). Firing was more suppressed after administering bupivacaine. Baseline responses for each group were evaluated using Friedman and Dunn's post hoc tests. The Mann–Whitney U test analyzed changes induced by anesthesia between the two groups. A Mann–Whitney U test FDR-adjusted p-value of < 0.05 indicated significant differences between the groups. #: significant differences vs. baseline. *: significant differences between bupivacaine and levobupivacaine.

levobupivacaine at 0.25% and 0.5% [18,19]. Indeed, all three of these doses are above the 95% effective dose ($ED_{95}$), so no substantial differences are expected [20,21]. At lower doses, however, levobupivacaine appears less effective than bupivacaine for blocking peripheral sensory fibers mediating tactile sensation, but roughly as effective for blocking nociceptive fibers.

The pricking pain sensation caused by stiff von Frey hairs is mainly transmitted by Aδ-fibers [22], while the heat pain sensation is transmitted by both Aδ- and C-fibers [23]. The tactile sensation from softer von Frey hairs is transmitted by Aβ-fibers [24]. The SDH contains neurons that respond differently to these inputs, including WDR neurons that respond to stimuli of different intensities, ranging from gentle to painful [25]. Conversely, LT neurons respond only to innocuous stimuli, mainly via Aβ-fibers [26]. In the present study, 0.025% and 0.0625%

levobupivacaine inhibited painful sensation but not tactile sensation in humans. In comparison, 0.05% levobupivacaine suppressed WDR neuron responses to noxious stimuli but not LT neuron responses to innocuous stimuli. These results suggest that levobupivacaine at low doses preferentially blocks Aδ- and C-fiber transmission, resulting in greater relative nociceptive-specificity. Uta and colleagues also reported that L-bupivacaine (levobupivacaine) preferentially inhibited the firing of nociceptive neurons, while D-bupivacaine blocked the firing of nociceptive and non-nociceptive neurons with roughly equal efficacy as evidenced by electrophysiological analysis of rat dorsal root ganglion neurons in vitro and spinal transmission in vivo [11].

Levobupivacaine and bupivacaine are optical isomers with identical dissociation constants, molecular weights, and liposolubility, key factors determining the activities of conventional local anesthetics [27]. Both compounds also target sodium channels. However, nine sodium channel subtypes have been identified (Nav 1.1–1.9), so differential fiber block may result from variations in channel subtype block efficacy combined with unique expression patterns among fiber types. Tetrodotoxin (TTX)-resistant Nav 1.8 and 1.9 are expressed at high levels in (nociceptive) Aδ- and C-fibers [28–30], and it has been reported that levobupivacaine has a higher affinity for TTX-resistant than TTX-sensitive sodium channels. In contrast, bupivacaine has shown no difference in affinity between tetrodotoxin-sensitive and tetrodotoxin-resistant sodium channels [11,31]. Thus, the differences in sensory block characteristics between bupivacaine and levobupivacaine may stem from differential blockade of TTX-sensitive and -resistant voltage-gated sodium channels.

These results illustrate the differences between the two optical isomers of local anesthetics. Notably, these differences were only evident at concentrations below 0.0625%, lower than those typically utilized in clinical practice [1,32]. In other words, the results observed in this study may be less noticeable when using the aforementioned local anesthetics at concentrations of 0.125% or higher in clinical settings. However, the results indicating that low-concentration levobupivacaine can achieve analgesia without tactile anesthesia suggest potential benefits in certain clinical situations. For instance, when rapid recovery from regional anesthesia is vital, as with day surgeries, patients may benefit from recovering their tactile sensation quickly while maintaining analgesia. Furthermore, because local anesthetic concentrations diminish with distance from the injection site, the tactile sensory block area induced by levobupivacaine may be smaller than bupivacaine at the equal doses and concentration. If the local anesthetic concentration can be reduced further by combining it with other analgesics, pain relief can be achieved without suppressing tactile sensation. Therefore, maintaining tactile sensation while selectively blocking pain could benefit various clinical scenarios. Further research is necessary to establish whether the differences between these two anesthetics can be observed in clinical practice settings.

The present study has several limitations. First, different noxious stimuli were used in human and rat experiments (thermal vs. pinch) as the thermal stimulator probe was too large for the rat hindpaw. Second, the vF hair stiffness was limited to 300 g and the heat stimulus to 47 °C in human experiments to prevent skin damage, which may have also differentially influenced the analgesic effects of bupivacaine and levobupivacaine. However, the recovery from levobupivacaine analgesia tended to be similar or slower than recovery from bupivacaine (Fig 3). Therefore, it is unlikely that the analgesic efficacy of levobupivacaine is inherently inferior to that of bupivacaine.

## Conclusions

We demonstrate that low-dose levobupivacaine can preferentially block the transmission of nociceptive information without affecting the transmission of innocuous tactile information,

while bupivacaine blocks both with roughly equal efficacy. Therefore, levobupivacaine may be a better choice for applications requiring pain suppression but maintenance of tactile sensation.

## Supporting information

**S1 Table. Carryover effect and period effect** .
(DOCX)

**S2 Table. Experiment 1 (human study)** .
(DOCX)

**S3 Table. Experiment 2 (animal study)** .
(DOCX)

## Acknowledgments

We would like to thank Dr. Masahiro Sakata (Research Planning & Drug Discovery Research, Research Laboratories, Maruho Co., LTD.) for technical assistance with the *in vivo* electrophysiological recordings.

## Author contributions

**Data curation:** Takashi Ishida.

**Formal analysis:** Takashi Ishida.

**Funding acquisition:** Akiyuki Sakamoto.

**Investigation:** Akiyuki Sakamoto, Satoshi Tanaka.

**Methodology:** Satoshi Tanaka.

**Project administration:** Akiyuki Sakamoto.

**Resources:** Akiyuki Sakamoto.

**Software:** Takashi Ishida.

**Validation:** Satoshi Tanaka.

**Visualization:** Akiyuki Sakamoto.

**Writing – original draft:** Akiyuki Sakamoto.

**Writing – review & editing:** Satoshi Tanaka, Takashi Ishida, Mikito Kawamata.

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
