## [Decision Letter · Decision Letter 0]

7 Nov 2024

PONE-D-24-22812Differences in sensory nerve block between levobupivacaine and bupivacaine at low concentrations in humans and animalsPLOS ONE

Dear Dr. Sakamoto,

Thank you for submitting your manuscript to PLOS ONE. After careful consideration, we feel that it has merit but does not fully meet PLOS ONE’s publication criteria as it currently stands. Therefore, we invite you to submit a revised version of the manuscript that addresses the points raised during the review process.

The manuscript has been evaluated by two reviewers, and their comments are available below.

The reviewers have raised a number of major concerns, specifically that the statistical methodology requires improving and that a reanalysis using mixed effects models for repeated measures should be performed. Could you please carefully revise the manuscript to address all comments raised?==============================

We look forward to receiving your revised manuscript.

Kind regards,

Johanna Pruller, Ph.D.

Staff Editor

PLOS ONE

Journal Requirements: When submitting your revision, we need you to address these additional requirements. 1. Please ensure that your manuscript meets PLOS ONE's style requirements, including those for file naming. The PLOS ONE style templates can be found at https://journals.plos.org/plosone/s/file?id=wjVg/PLOSOne_formatting_sample_main_body.pdf and https://journals.plos.org/plosone/s/file?id=ba62/PLOSOne_formatting_sample_title_authors_affiliations.pdf 2. To comply with PLOS ONE submissions requirements, in your Methods section, please provide additional information regarding the experiments involving animals and ensure you have included details on (1) methods of sacrifice, (2) methods of anesthesia and/or analgesia, and (3) efforts to alleviate suffering. 3. Thank you for stating the following financial disclosure: "This work was financially supported by the Japan Society for the Promotion of Science Grants-in-Aid for Scientific Research Grant Number 23791696. " Please state what role the funders took in the study.  If the funders had no role, please state: ""The funders had no role in study design, data collection and analysis, decision to publish, or preparation of the manuscript."" If this statement is not correct you must amend it as needed. Please include this amended Role of Funder statement in your cover letter; we will change the online submission form on your behalf. 4. In the online submission form, you indicated that "The datasets used and/or analysed during the current study are available from the corresponding author on reasonable request." All PLOS journals now require all data underlying the findings described in their manuscript to be freely available to other researchers, either 1. In a public repository, 2. Within the manuscript itself, or 3. Uploaded as supplementary information.This policy applies to all data except where public deposition would breach compliance with the protocol approved by your research ethics board. If your data cannot be made publicly available for ethical or legal reasons (e.g., public availability would compromise patient privacy), please explain your reasons on resubmission and your exemption request will be escalated for approval. 5. Please include captions for your Supporting Information files at the end of your manuscript, and update any in-text citations to match accordingly. Please see our Supporting Information guidelines for more information: http://journals.plos.org/plosone/s/supporting-information.

Reviewers' comments:

Reviewer's Responses to Questions

**Comments to the Author**

1. Is the manuscript technically sound, and do the data support the conclusions?

Reviewer #1: Yes

Reviewer #2: Yes

2. Has the statistical analysis been performed appropriately and rigorously? 

Reviewer #1: No

Reviewer #2: I Don't Know

3. Have the authors made all data underlying the findings in their manuscript fully available?

Reviewer #1: Yes

Reviewer #2: Yes

4. Is the manuscript presented in an intelligible fashion and written in standard English?

Reviewer #1: Yes

Reviewer #2: Yes

5. Review Comments to the Author

Reviewer #1: A randomized crossover clinical trial was conducted which aimed to compare the sensory blocking characteristics of levobupivacaine to bupivacaine in human and animal models. The results are unclear.

Major revisions:

Superior statistical testing methods are available for analyzing repeated measures data. Provide a comprehensive reanalysis using mixed effects models for repeated measures.

Minor revisions:

1- Verify that the data was normally distributed prior to applying either the t-test or paired t-tests.

2- Line 214: State the statistical testing method which achieves 80% power.

3- Line 221: Provide more details about the sample size calculation for Experiment 2. The power calculation should include: sample size, alpha level (indicating one or two-sided), minimal detectable difference and statistical testing method.

4- Cite the statistical software used for the analysis.

5- Table 1: In addition to the frequencies, indicate the percentage male and female.

6- The standard statistical term for average is mean.

Reviewer #2: This rigorous and well-written manuscript documents and demonstrates a differential action of levopuvicaine compared to the racemic mixture at low dosage on the basis of converging results from original animal experiments and a clinical trial by subcutaneous administration.

However, the modalities of a possible application in anesthesia remain to be defined: effective dose, site of application.

Some typos or expressions can be improved:

Abstract l.26: A space is missing "characteristicsis"

Introduction l.57: probably rather “compared to” than “than”

Experiment (1) l.107: omit years old

Experiment (2) l.195-198: the sentences ending with "in order" seem incomplete

Fig 1: the arrows are ambiguous. Prefer distinct lines between the two phases or reverse the blocks (it looks like the levobupivacaine group is still receiving levobupivacaine in phase 2)

6. PLOS authors have the option to publish the peer review history of their article (what does this mean? ). If published, this will include your full peer review and any attached files.

**Do you want your identity to be public for this peer review?** For information about this choice, including consent withdrawal, please see our Privacy Policy .

Reviewer #1: No

Reviewer #2: No

---

## [Author Response · Author response to Decision Letter 1]

12 Dec 2024

We would like to thank the academic editor and reviewers for carefully reading our manuscript and for the insightful comments. The comments led to an improvement of the work. Detailed responses to the reviewers are shown below. The comments from the reviewers are shown in black and our replies are shown in red. In this revised version, changes to our manuscript within the document were all highlighted by using red-colored text (with underlining) in the file named “Revised Manuscript with Track Changes”.

Reply to Reviewer #1

Reviewer #1: A randomized crossover clinical trial was conducted which aimed to compare the sensory blocking characteristics of levobupivacaine to bupivacaine in human and animal models. The results are unclear.

Major revisions:

Superior statistical testing methods are available for analyzing repeated measures data. Provide a comprehensive reanalysis using mixed effects models for repeated measures.

Thank you for highlighting this important issue. We concur that statistical analysis is vital in research. In the original version, we did not check for the normality of the data distribution. At present, we have assessed the data distribution using the Shapiro–Wilk test. Consequently, we could not confirm the normality of certain data distributions. Therefore, we employed nonparametric methods for a reanalysis of our findings data.

The statistical methods described below were used; however, the results of the original version of the statistical analysis were the same. Therefore, no significant changes were needed for the discussion or conclusion.

Minor revisions:

1- Verify that the data was normally distributed prior to applying either the t-test or paired t-tests.

As mentioned above, the Shapiro–Wilk test could not confirm the normality of data distribution for some of data; therefore, nonparametric analyses were conducted.

We analyzed the data using the Friedman test followed by Dunn’s post hoc test for within-group comparisons, and Mann–Whitney U test for between-group comparisons. The p-values obtained from the Mann–Whitney U test were corrected for multiple testing using the Benjamini and Hochberg false discovery rate (FDR) method (Benjamini et al; Glickman et al). We set the FDR threshold at 0.05. In the Results section, we reported the FDR-adjusted p-value. A significant difference was regarded as an FDR-adjusted p-value <0.05.

To summarize, in statistics, multiple testing leads to a higher likelihood of type I errors when tests are conducted multiple times. Implementing the FDR helps mitigate type I errors effectively proportion.

Benjamini Y, Hochberg Y. Controlling the false discovery rate: a practical and powerful approach to multiple testing. J R Stat Soc Ser B. 1995;57:289-300.

Glickman ME, Rao SR, Schultz MR. False discovery rate control is a recommended alternative to Bonferroni-type adjustments in health studies. J Clin Epidemiol. 2014;67:850-857.

2- Line 214: State the statistical testing method which achieves 80% power.

In an animal study by Uta et al., bupivacaine and levobupivacaine showed a considerable difference in their inhibition rates to non-noxious stimuli, which is illustrated in a figure displaying the mean and standard deviation. While planning our research project, some of Uta et al.’s findings were shared at a conference in Japan. The difference in means exceeded twice the standard deviation. Consequently, we established the effect size at 2 and computed the sample size for our study. An 80% power is a widely accepted benchmark in research. Conducting under or overpowered studies is unethical and a financial waste; hence, the recommendation is for a nominal statistical power of 80% (Nakagawa S et al.).

Uta D, Koga K, Furue H, Imoto K, Yoshimura M. L-bupivacaine inhibition of nociceptive transmission in rat peripheral and dorsal horn neurons. Anesthesiology. 2021;134:88-102.

Nakagawa S, Lagisz M, Yang Y, Drobniak SM. Finding the right power balance: better study design and collaboration can reduce dependence on statistical power. PLoS Biol. 2024;22:e3002423.

3- Line 221: Provide more details about the sample size calculation for Experiment 2. The power calculation should include: sample size, alpha level (indicating one or two-sided), minimal detectable difference and statistical testing method.

In our animal study (Experiment 2), the sample size was established based on a prior study with a similar design (Sagar et al.) and a power analysis conducted using G*Power software. We calculated the sample size using G-Power 3.1, considering a type I error rate of α = 0.05, power level of 80% (1 − β) = 0.80, an effect size of 2, and two-sided test. Therefore, the minimum required sample size was n = 5 for each group in our animal study.

Although expanding the sample size can reduce type II errors, it will raise the project costs and prolong the timeline for completing the research. Furthermore, excessive sampling could lead to ethical concerns (Serdar CC et al.). The relationships between power, alpha values, sample size, and effect size are intricately connected. As previously stated, we have meticulously established a “cost-effective sample size” for our study.

Sagar DR, Kendall DA, Chapman V. Inhibition of fatty acid amide hydrolase produces PPAR-alpha-mediated analgesia in a rat model of inflammatory pain. Br J Pharmacol. 2008;155: 1297-1306.

Serdar CC, Cihan M, Yücel D, Serdar MA. Sample size, power and effect size revisited: simplified and practical approaches in pre-clinical, clinical and laboratory studies. Biochem Med (Zagreb). 2021 ;31:010502.

4- Cite the statistical software used for the analysis.

We performed statistical analysis on our data using Prism 9.0 (GraphPad Software, San Diego, CA). The statistical analyses section has been updated as follows:

“We used G-Power 3.1 to determine the sample size, with a type I error rate (α) of 0.05, power value of 80% (1 − β = 0.80), an effect size of 2, and two-sided test. The minimum required group size for our human volunteer study was n = 5. Given the crossover design and potential dropouts, we increased the sample size to six. Our animal study established the sample size referring to prior studies that employed a similar design [16] and power analysis with G*Power software. Following the Shapiro–Wilk normality test, data were represented as mean ± standard deviation or median (25%, 75% interquartile range), depending on appropriateness. For within-group comparisons, Friedman and Dunn’s post hoc tests were utilized, with p < 0.05 considered statistically significant. The Mann–Whitney U test was employed for group comparisons. Multiple testing indicates the increased risk of type I error while repeatedly conducting multiple statistical tests. The false discovery rate (FDR) correction aims to minimize the occurrence of type I errors at a manageable proportion [17]. The FDR-adjusted p-value was computed using this method, with an FDR-adjusted p-value <0.05 deemed significant.”

5- Table 1: In addition to the frequencies, indicate the percentage male and female.

The following changes were made to Table 1.

male female total

N (%) 15 (83.3) 3 (16.7) 18 (100)

6- The standard statistical term for average is mean.

The term “average” has been changed to “mean” in the revised version of this manuscript.

Reply to Reviewer #2

Reviewer #2: This rigorous and well-written manuscript documents and demonstrates a differential action of levobupivacaine compared to the racemic mixture at low dosage on the basis of converging results from original animal experiments and a clinical trial by subcutaneous administration.

However, the modalities of a possible application in anesthesia remain to be defined: effective dose, site of application.

As the reviewer pointed, discussing the study implications is crucial for clinical practice. Our research highlighted the differences between the two optical isomers of local anesthetics; however, these differences were only detected at concentrations below 0.0625%, lower than those typically used in practice. The findings may be less apparent when local anesthetics are administered at concentrations ≥0.125% in clinical environments. Nonetheless, scenarios in which swift recovery from regional anesthesia is vital, such as in day surgery, it is advantageous for patients to regain tactile sensation quickly while still providing pain relief. Furthermore, because the concentration of local anesthetic diminishes with increasing distance from the injection site, the tactile sensory block offered by levobupivacaine could be more limited than bupivacaine at equal doses and concentrations. Should the concentration of local anesthetics be reduced further in conjunction with other analgesics, there may be an opportunity to relieve pain without compromising sensory function using local anesthetics.

We therefore modified the discussion section as follows:

“These results illustrate the differences between the two optical isomers of local anesthetics. Notably, these differences were only evident at concentrations below 0.0625%, lower than those typically utilized in clinical practice [1, 33]. In other words, the results observed in this study may be less noticeable when using the aforementioned local anesthetics at concentrations of 0.125% or higher in clinical settings. However, the results indicating that low-concentration levobupivacaine can achieve analgesia without tactile anesthesia suggest potential benefits in certain clinical situations. For instance, when rapid recovery from regional anesthesia is vital, as with day surgeries, patients may benefit from recovering their tactile sensation quickly while maintaining analgesia. Furthermore, because local anesthetic concentrations diminish with distance from the injection site, the tactile sensory block area induced by levobupivacaine may be smaller than bupivacaine at the equal doses and concentration. If the local anesthetic concentration can be reduced further by combining it with other analgesics, pain relief can be achieved without suppressing tactile sensation. Therefore, maintaining tactile sensation while selectively blocking pain could benefit various clinical scenarios. Further research is necessary to establish whether the differences between these two anesthetics can be observed in clinical practice settings.”

Some typos or expressions can be improved:

Abstract l.26: A space is missing "characteristicsis"

Thank you very much. The correction was made following your suggestion.

Introduction l.57: probably rather “compared to” than “than”

We made the suggested correction.

Experiment (1) l.107: omit years old

This correction was made.

Experiment (2) l.195-198: the sentences ending with "in order" seem incomplete

We have changed “in order” to “sequentially.”

Fig 1: the arrows are ambiguous. Prefer distinct lines between the two phases or reverse the blocks (it looks like the levobupivacaine group is still receiving levobupivacaine in phase 2)

In response, we have updated Figure 1 to clarify the meaning of the various arrows.

The following is a modified diagram.

Fig. 1

The statistical analysis has been redone, so the significant differences in data have changed somewhat. An overview is provided below. There was no significant difference in the original version, but the areas where a significant difference was observed this time are indicated with a red circle. The areas with a significant difference in the original version but no significant difference this time are indicated in green.

Fig. 3

Fig. 4

Fig. 5

Fig. 6

---

## [Decision Letter · Decision Letter 1]

7 Jan 2025

PONE-D-24-22812R1Differences in sensory nerve block between levobupivacaine and bupivacaine at low concentrations in humans and animalsPLOS ONE

Dear Dr. Sakamoto,

Thank you for submitting your manuscript to PLOS ONE. After careful consideration, we feel that it has merit but does not fully meet PLOS ONE’s publication criteria as it currently stands. Therefore, we invite you to submit a revised version of the manuscript that addresses the points raised during the review process.

We look forward to receiving your revised manuscript.

Kind regards,

Anjan Khadka, MBBS, MD

Academic Editor

PLOS ONE

Journal Requirements:

Reviewers' comments:

Reviewer's Responses to Questions

**Comments to the Author**

1. If the authors have adequately addressed your comments raised in a previous round of review and you feel that this manuscript is now acceptable for publication, you may indicate that here to bypass the “Comments to the Author” section, enter your conflict of interest statement in the “Confidential to Editor” section, and submit your "Accept" recommendation.

Reviewer #1: (No Response)

Reviewer #2: All comments have been addressed

2. Is the manuscript technically sound, and do the data support the conclusions?

Reviewer #1: Yes

Reviewer #2: Yes

3. Has the statistical analysis been performed appropriately and rigorously? 

Reviewer #1: Yes

Reviewer #2: I Don't Know

4. Have the authors made all data underlying the findings in their manuscript fully available?

Reviewer #1: Yes

Reviewer #2: Yes

5. Is the manuscript presented in an intelligible fashion and written in standard English?

Reviewer #1: Yes

Reviewer #2: Yes

6. Review Comments to the Author

Reviewer #1: Minor revisions:

Line 212: State the statistical testing method which achieves 80% power. Examples of testing methods are t-test, ANOVA, and Fisher's Exact test etc.

Reviewer #2: Thank you very much for the additions to your manuscript and for your responses to my comments.

7. PLOS authors have the option to publish the peer review history of their article (what does this mean? ). If published, this will include your full peer review and any attached files.

**Do you want your identity to be public for this peer review?** For information about this choice, including consent withdrawal, please see our Privacy Policy .

Reviewer #1: No

Reviewer #2: No

---

## [Author Response · Author response to Decision Letter 2]

15 Jan 2025

Reply to academic editor

On January 12, 2025, we re-evaluated all references 1 to 32 of our manuscript (PONE-D-24-22812) using the literature search system “Medline” and we confirmed that there were no retracted papers. Therefore, no deletions or changes to the references are necessary.

Reply to Reviewer #1

Reviewer #1: Minor revisions:

Line 212: State the statistical testing method which achieves 80% power. Examples of testing methods are t-test, ANOVA, and Fisher's Exact test etc.

Thank you for pointing this out.

We used the t-test for our 80% power statistic. Therefore, we have rewritten the text of Line 212 as follows.

Line 212: We used G-Power 3.1 to determine the sample size, with a type I error rate (α) of 0.05, power value of 80% (1 − β = 0.80), an effect size of 2, and two-sided t-test.

Before the start of the study, we anticipated that the data would be normally distributed and considered using a t-test for statistical analysis. However, since the data did not exhibit normality, we conducted statistical analysis using the Mann-Whitney U test. With a type I error rate of 0.05, a power value of 80% (1 − β = 0.80), and an effect size of 2, the required sample size for the Mann-Whitney U test was calculated to be n = 6 using G-Power 3.1. In any case, we believe that this study was conducted with an appropriate sample size.

---

## [Editor Report · Decision Letter 2]

17 Jan 2025

Differences in sensory nerve block between levobupivacaine and bupivacaine at low concentrations in humans and animals

PONE-D-24-22812R2

Dear Dr. Sakamoto,

We’re pleased to inform you that your manuscript has been judged scientifically suitable for publication and will be formally accepted for publication once it meets all outstanding technical requirements.

Kind regards,

Anjan Khadka, MBBS, MD

Academic Editor

PLOS ONE
---

## [Editor Report · Acceptance letter]

PONE-D-24-22812R2

PLOS ONE

Dear Dr. Sakamoto,

I'm pleased to inform you that your manuscript has been deemed suitable for publication in PLOS ONE. Congratulations! Your manuscript is now being handed over to our production team.

Kind regards,

on behalf of

Dr. Anjan Khadka

Academic Editor

PLOS ONE